# Sulfidogenic Bioreactor-Mediated Formation of ZnS Nanoparticles with Antimicrobial and Photocatalytic Activity

**DOI:** 10.3390/nano13050935

**Published:** 2023-03-04

**Authors:** Aileen Segura, Araceli Rodriguez, Pedro Hernández, Hector Pesenti, Jacobo Hernández-Montelongo, Antonio Arranz, Noelia Benito, José Bitencourt, Luis Vergara-González, Iván Nancucheo, Gonzalo Recio-Sánchez

**Affiliations:** 1Facultad de Ingeniería, Arquitectura y Diseño, Universidad San Sebastián, Concepción 4030000, Chile; 2Núcleo de Investigación en Bioproductos y Materiales Avanzados, Universidad Católica de Temuco, Temuco 4780000, Chile; 3Departamento de Física Aplicada, Universidad Autónoma de Madrid, Cantoblanco, 28049 Madrid, Spain; 4Departamento de Física, Universidad de Concepción, Concepción 4030000, Chile; 5Instituto Tecnológico Vale, Belém 66055-090, Brazil; 6Facultad de Medicina y Ciencia, Universidad San Sebastián, Concepción 4030000, Chile

**Keywords:** sulfidogenic bioreactor, ZnS nanoparticles, photocatalytic activity, antibacterial activity, organic dye degradation

## Abstract

The use of sulfidogenic bioreactors is a biotechnology trend to recover valuable metals such as copper and zinc as sulfide biominerals from mine-impacted waters. In the present work, ZnS nanoparticles were produced using “green” H_2_S gas generated by a sulfidogenic bioreactor. ZnS nanoparticles were physico-chemically characterized by UV-vis and fluorescence spectroscopy, TEM, XRD and XPS. The experimental results showed spherical-like shape nanoparticles with principal zinc-blende crystalline structure, a semiconductor character with an optical band gap around 3.73 eV, and fluorescence emission in the UV-visible range. In addition, the photocatalytic activity on the degradation of organic dyes in water, as well as bactericidal properties against several bacterial strains, were studied. ZnS nanoparticles were able to degrade methylene blue and rhodamine in water under UV radiation, and also showed high antibacterial activity against different bacterial strains including *Escherichia coli* and *Staphylococcus aureus.* The results open the way to obtain valorous ZnS nanoparticles from the use of dissimilatory reduction of sulfate using a sulfidogenic bioreactor.

## 1. Introduction

Mine-impacted waters (MIW) can contain elevated concentration of transition metals and sulfate, which cause severe environmental issues, though given concerns regarding future availability of metals, their recovery from MIW can offer represent a secondary source of the materials. Currently, 30% of global production of Zn derives from recycling, and demand is expected to increase up to 130% by the end of this century. Therefore, the recovery of metals from secondary source will grow continuously [1]. A promising biotechnology approach to recover Zn from mine waters is based on the ability of sulfate -reducing bacteria (SRB) in which hydrogen sulfide generated from the reduction of sulfate can selectively precipitate with zinc as zinc sulfide (ZnS) [2]. Recently, biogenic H_2_S produced by a sulfate-reducing bioreactor has been used to demonstrate the formation of valuable CuS nanoparticles [3].

ZnS nanostructures have been researched over the last years due to their excellent optoelectronic properties. ZnS is a well-known semiconductor material with an electronic band gap around 3.7 eV, depending on crystalline structure, which can be selected between cubic zincblende and wurtzite structures [4,5]. In addition, ZnS has high stability, a high diffractive index and transmittance in the visible range. Thus, ZnS nanostructures have been used for several optoelectronic applications such as light-emitting diodes, photonic, non-optical devices and sensors, among others [6,7]. Moreover, they show photocatalytic activity for the degradation of persistent organic dyes in water, such as methylene blue (MB), rhodamine-B (RhB) and methylene orange, and antibacterial activity, and may be used for other kind of applications including wastewater treatment and antimicrobial surfaces [8,9,10]

Different physico-chemical methods for the synthesis of ZnS nanostructures have been proposed in recent years. Precipitation and thermolysis methods are common techniques to synthesize ZnS nanostructures and doped ZnS nanostructures [11]. By an appropriate choice of precursors, these methods allow control of the main physico-chemical properties of the resulting ZnS nanoparticles, such as the size, shape and crystallinity nature [12,13,14]. Similarly, other chemical methods, such as hydrothermal and sonochemical techniques, have also allowed the synthesis of monodispersed ZnS nanoparticles with different sizes and shapes by changing the main parameters in the process, including pH, or precursors, and these techniques even allow in-situ synthesis of ZnS nanoparticles on different substrates [15,16,17,18,19]. Other chemical synthesis methods include microwave irradiation and aerosol microdroplets [20,21]. Green synthesis using plant extracts and bacteria as precursor and stabilizing agents has also been studied as an eco-friendly alternative avoiding the use of chemical agents [22,23]. In this work, we studied the synthesis of ZnS nanoparticles by using a sulfate reducing bioreactor. The use of bioreactors and supply of the carbon source for sulfate reduction can increase the operational cost of remediation of MIW. Thus, the challenge of this biotechnology is to allow the simultaneously removal of the sulfate present in the MIW and recover the metal content as pure valuable metal sulfides that can be used in further applications [3].

In the present study, biogenic H_2_S produced from an upflow biofilm continuous sulfidogenic bioreactor was delivered from the off-gas to an aqueous solution of ZnSO_4_, promoting the formation of ZnS nanoparticles. The main physico-chemical properties of ZnS nanoparticles were studied by TEM, XRD, UV-vis, fluorescence spectroscopy and XPS. In addition, their photocatalytic activity on the degradation of MB and RhB under visible UV light and their antibacterial activity against several bacterial strains were analyzed. The outputs of this work show the multifunctional potential application of ZnS nanoparticles synthetized from biogenic H_2_S, opening the door to the implementation of sulfidogenic bioreactor technology to treat sulfate wastewater and obtain valuable nanoparticle precipitates.

## 2. Materials and Methods

### 2.1. Synthesis of ZnS Nanoparticles

ZnS nanoparticles were synthesized by delivering a H_2_S gas stream produced by a sulfidogenic bioreactor to an aqueous solution of 5 mM ZnSO_4_ (Merck Millipore, Burlington, MA, USA). A 2.3 L working volume sulfidogenic bioreactor (Fermac 200; Electrolab Biotech, Tewkesbury, United Kingdom) was operated as an upflow biofilm continuous sulfidogenic bioreactor at pH 7.0, as described elsewhere [24], housing sulfidogenic bacteria dominated by the SRB genera of *Desulfomicrobium*, *Desulfobacterium* and *Desulfovibrio* that were obtained from different ponds in Salar de Huasco, Chile [25] having blackened sediments. Enrichments were grown on porous beads of recycled glass of 1–2 mm diameter. The bioreactor was fed 5 mM lactate, 1000 ppm sulfate, autotrophic basal salt (ABS; [26]), trace elements and yeast extract at pH of 6.5. The H_2_S gas was removed by a stream of oxygen-free nitrogen (OFN) at a flux to 100 mL·min^−1^ and transferred to an off-line vessel containing 50 mL of an aqueous solution of ZnSO_4_. After the zinc solution was sparged with gas for 60 min, the precipitate formed (ZnS nanoparticles) was collected from the solution, dried at 30 °C for 24 h and stored for further use.

### 2.2. Characterization Techniques

The morphology and size of the synthesized nanoparticles were observed with a transmission electron microscope (Hitachi HT7700, 120 kV). A spectrophotometer (Epoch Biotek Santa Clara, CA, USA) was used to record UV-Vis absorption spectrum of ZnS nanoparticles. An aqueous solution of 1 mg/mL of ZnS nanoparticles was prepared by stirring for 1 h, and 200 μL of the solution was deposited on a 96-well UV-vis plate. The absorption spectrum was recorded at an interval of 1 nm. The fluorescence spectrum was obtained using a fluorescence spectrophotometer (SynergyH1 Biotek Santa Clara, CA, USA) with an interval of 1 nm. The X-ray diffraction pattern was recorded with a Rigaku X-Ray diffractometer Smartlab model (Rigaku Corporation, Tokyo, Japan). Ni-filtered K_α_ Cu radiation was used with a 0.5° divergence slit, a solid-state detector D/teX Ultra 250 model and a goniometer Theta-Theta Bragg-Brentano geometry alignment with LaB6 SMR NIST 660c. X-ray photoelectron spectroscopy (XPS) was carried out in an ultra-high vacuum chamber using a hemispherical analyzer (SPECS Phoibos 100MDC-5, Berlin, Germany) and Mg Kα (1253.6 eV) radiation from a twin anode (Al–Mg) X-ray source operated at 300 W. Before the spectral analysis, the binding energy (BE) was calibrated using the C 1s peak with adventitious carbon contribution centered at 284.5 eV.

### 2.3. Photocatalytic Degradation Test

The degradation of organic dyes MB and RhB were investigated by adding 2 mL of H_2_O_2_ (60% wt) to 18 mL of aqueous solution of organic dye (3.1 × 10^−5^ M). After 5 min of magnetic stirring in dark, 20 mg of ZnS photocatalyst were added to the solution and the resulting solution was illuminated with a UV source at 1 W/cm^2^ of power density. To record the degradation rate of each organic dye, 200 µL of solution was taken from the solution and the UV-Vis absorption spectrum was measured. At the same time, the control of degradation of each organic dye was carried out in the same condition, but without adding the ZnS photocatalyst.

### 2.4. Antibacterial Assays

Antibacterial activity of ZnS nanoparticles was measured by the agar diffusion method against different bacterial strains: Gram-positive (*Staphylococcus aureus*, *Bacillus* sp., *Enterococcus faecalis* and *Staphylococcus saprophyticus*) and Gram-negative (*Enterobacter cloacae*, *Klebsiella pneumoniae*, *Pseudomonas aeruginosa* and *Escherichia coli*). Mueller-Hilton agar was used for the diffusion method test. The controlled concentration of each bacterial strain was adjusted to a 0.5 McFarland standard inoculated onto solid media under aseptic conditions. Aliquots of 10 µL of an aqueous solution of ZnS nanoparticles with two different concentrations (20 mg/mL and 40 mg/mL) were poured onto the solid media surface and left to dry for one hour. The inoculated plates were incubated for 24 h at 35 °C, then the inhibition zones around the drops were measured from photographs of the plates using ImageJ software. Each bactericidal assay was done in triplicate, and statistical analyses of the inhibition zone were performed for each concentration.

## 3. Results and Discussion

### 3.1. ZnS NPs Characterization

Figure 1a shows a TEM image of ZnS nanoparticles bioproduced after delivering the H_2_S generated by the sulfidogenic bioreactor to an aqueous solution of ZnSO_4_. It can be seen that ZnS nanoparticles have a wide size dispersion and tendency to agglomerate, suggesting low stability with poor applicability [3]. Figure 1b shows the size histogram of the ZnS nanoparticles elaborated from TEM images using ImageJ software, with a Gaussian distribution and a mean diameter of 107 ± 20 nm. Inset Figure 1b shows a zoom of one ZnS nanoparticles, where the spherical-like shape of the bioproduced NPs can be observed. Under these conditions, the average size of the final ZnS nanoparticles was slightly higher than that obtained from other synthesis methods such as hydrothermal or solvothermal techniques, where ZnS nanoparticles were selected from a few nanometers of diameter [15,16,17].

Figure 2a shows the UV-vis absorption spectrum of ZnS nanoparticles. A broad absorption band from 250 to 350 nm of wavelength can be observed with the maximum absorption peak at 335 nm. The absorbance decays at higher wavelengths. The shape of the absorption spectrum is blue-shifted with respect to the reported for ZnS macroparticles, due to the quantum confinement effects [27]. These effects have also been reported for other kind of ZnS nanoparticles. ZnS nanoparticles with smaller diameters and synthesized using a layer-by-layer technique in a polymeric matrix showed similar absorption bands in the same range, presenting blue shifts depending on the size of the nanoparticle [28]. Similarly, ZnS nanoparticles formed by a co-precipitation method using Schiff base exhibited absorbance spectrum with similar effects [29]. On the other hand, monodispersed ZnS nanoparticles show defined absorption peaks correlated to the quantum effects, as was reported with ZnS nanoparticles synthesized via chemical in situ techniques [30,31]. Moreover, the quantum effects may be influenced by the morphology of the ZnS nanoparticles and their stability [32]. In this work, the not-well defined absorption band can be attributed to nanoparticles morphology characterized by sizes higher than 100 nm, low stability and tendency to agglomerate [31,32]. From UV-vis spectra, the optical band gap was calculated from a Tauc plot (Figure 2b). First the absorption coefficient (*α*) was shifted such that the lowest value was set to zero to account for any wavelength-independent reflectance and scattering [33,34]. Then, *(αhν)^2^* was plotted as a function of photon energy *hν*, and the band gap was obtained by extrapolating the linear region to the horizontal axis. In this case, the obtained band gap was 3.73 eV. This value can be modified depending on several factors such as the crystallinity, the shape or the stability of the NPs, among others. ZnS nanoparticles with lower main diameters usually showing higher energy band gap values [29,30]. ZnS nanoparticles synthesized by the chemical route showed few different band gaps, ranging between 3.98 and 3.87 eV depending on the reaction time [29]. On the other hand, ZnS nanostructures with diverse morphologies can modify the band gaps from 3.5 to 3.8 eV [32]. The values obtained in this work match those of other kind of nanoparticles with mixed crystalline structures, but are closed to the expected from bulk ZnS due to the large size of the ZnS nanoparticles. [32,35].

Photoluminescence is an important optical parameter for the development of optoelectronic and sensing applications, and is influenced by the shape, size surface and effect energetic states, among other factors. Figure 3 shows the fluorescence spectra of ZnS at different excitation wavelengths. For excitation wavelengths with higher energies than the band gap (280 nm, 300 nm and 320 nm), an emission peak can be observed centered around 355 nm and 365 nm. These emission peaks may be related to the recombination of electron-hole pairs after photon absorption, being similar to previous reports [36,37]. Moreover, the broad peak can be associated with the wide size distribution of the ZnS nanoparticles, since the quantum effects of nanometer size can cause different wavelength emissions. In addition, a small red shift of these peaks was observed as the excitation wavelength increased. That obtained at an excitation wavelength of 280 nm was centered at 356 nm, and the peak was shifted to 360 nm and 366 nm for 300 nm and 320 nm of excitation wavelength, respectively. This shift can be related to selective excitation of the surface states in the nanoparticles [38]. For higher excitation wavelengths (340 nm and 360 nm), this main peak disappears. However, others smooth peaks can be observed (inset Figure 3) at 427 nm and 445 nm. These peaks may be related to defect states or interstitial impurities [30,36]. Nonetheless, the main fluorescence emission occurs over the UV range, and is difficult to use for optoelectronic or sensing applications [31]. Other well defined ZnS nanoparticles synthesized by chemical methods could shift the photoluminescence emission to the visible wavelength range [29]. Balayeva N. et al. observed an emission band centered at 430 nm for ZnS nanoparticles with a main diameter of 10 nm [28], and Mamiyev Z. et al. reported emission bands ranging from 401 nm to 418 nm depending on the main size of each nanoparticle [29]. In addition, Schiff base coordinated ZnS nanoparticles excited at 340 nm exhibited a broad photoluminescence emission centered at 432 nm [30]. These effects were explained by electron-hole recombination with energy states within the band gap. These energy states may be associated with surface defects, defect states provoked by S and Zn vacancies, and the size dispersion of the nanoparticles [29].

Figure 4 presents the X-ray diffraction pattern of the ZnS nanoparticles. The main diffraction peaks at 28.9°, 48.2° and 57.0° 2θ perfectly match the crystallographic planes (111), (220) and (311), respectively, related to the zincblende crystalline structure (cubic, space group F-43m). The average lattice parameter of the structure obtained from the data was a = 0.5409 nm, according to ICDD PDF Card: 04-012-0803. However, other minor diffraction peaks appeared at 34°, 36.2° and 47.6° 2θ, which can be related to the crystallographic planes (002), (101) and (102) of the wurtzite crystalline structure (hexagonal, space group P63mc). Moreover, the main peaks at 28.9° and 57° seem to lose symmetry since the peaks from both structures could be overlapped. The average lattice parameters of wurtzite crystalline domains are a = 0.3822 nm and c = 0.6375 nm, according to ICDD PDF Card: 00-036-1450.

Average nanocrystalline size (*D*) at all diffraction peaks was calculated following Debye-Scherrer:(1)$$D=\frac{k\cdot \lambda }{\beta cos\theta }$$

With the shape factor *k* as 0.9, λ as 0.1541 nm, and *β* and *θ* the width half maximum of each diffraction peak in radian and the diffraction angle, respectively. The obtained average crystalline domain was *D* = 4.7 ± 0.4 nm. The relatively small magnitude of the crystalline size in comparison with the size of the ZnS nanoparticles obtained by TEM (107 ± 20 nm) confirmed the mixture of both crystalline structures inside the NPs, zincblende being the predominant structure. This result is in agreement with the calculated optical band gap, which matches the values previously reported for ZnS nanoparticles with mixed crystalline structures [32,35].

To obtain deeper information about the chemical composition of ZnS nanoparticles, XPS measurements were performed. Figure 5a shows high-resolution XPS spectra of the Zn 2p doublet core level. In this spectrum, Zn 2p_3/2_ and Zn 2p_1/2_ peaks were observed at 1021.8 and 1044.9 eV respectively, with a spin-orbit splitting of 23.1 eV. The binding energy of these peaks is related to Zn^2+^, as reported by Madhusudan et al. [39]. Specifically, the binding energy observed for the Zn 2p_3/2_ is characteristic of Zn-S bonds [39,40]. Moreover, Figure 5b shows the Zn LMM Auger transition at a kinetic energy of 989.6 eV, in good agreement with values reported for ZnS [40]. In addition, the modified Auger parameter *α’* can be calculated as an unequivocal indicator of the chemical state of the NPs, since this parameter is not affected by surface charge. The modified Auger parameter was calculated as *α’* =*E_kin_* (Zn LMM) + *EB* (Zn 2p_3/2_) = 2011.4 eV, in good agreement with that previously reported for ZnS [40]. Figure 5c shows the S 2p doublet core level. In this spectrum, two different contributions can be clearly differentiated. The most intense one, with S 2p_3/2_ and S 2p_1/2_ components at 161.2 eV and 162.4 eV, respectively, is related to ZnS species [41,42] and the other one, at 168.2 eV and 169.4 eV, respectively, indicates the presence of sulfate species which can be attributed to remaining MIW [3]. These results agree with those obtained for ZnS nanoparticles synthetized by chemical [13,39] and hydrothermal methods [17,41].

### 3.2. Photocatalytic Activity

To study the photocatalytic activity of ZnS nanoparticles on the degradation of organic dyes MB and RhB in water, 2 mL of H_2_O_2_ was added to 18 mL of aqueous organic dye solution. After that, 20 mg of ZnS photocatalyst were added to the solution, and it was irradiated under UV light. H_2_O_2_ was included to promote the formation of oxidizing species. As a control, similar solutions were analyzed without adding ZnS nanoparticles. Figure 6a,b shows the degradation of MB and RhB, respectively, under UV light. It can be observed that only the 12% and 14% of MB and RhB, respectively, were removed after 180 min without the addition of ZnS nanoparticles. When 20 mg of ZnS were included into the solutions, the photocatalytic activity of ZnS nanoparticles allowed degradation of 58% and 68% of MB and RhB, respectively, at the same time. According to previous reports, ZnS nanoparticles can promote the formation of oxidizing species under UV light. The UV light can generate electron-hole pairs which may migrate to the surface of ZnS nanoparticles if they are not recombined. These electron-hole pairs can react with OH^-^ radicals and H_2_O_2_ molecules to promote the formation of ^•^OH and ^•^$O_{2}^{-}$ oxidizing species [10]. These oxidizing species play a key role in the degradation of the organic dyes since they can react with the main bond of these macromolecules.

To obtain deeper information about the mechanism of the photodegradation process, experiments were performed using scavengers. Propanol and EDTA were used as scavengers of ^•^OH and h^+^, respectively. The results of the percentage degradation of each organic dye are shown in Figure 7. In the presence of propanol, the percentage degradation of both dyes was clearly reduced, suggesting that ^•^OH species play an essential role in the photodegradation of both dyes, similar to previously reported works [43,44]. However, in the presence of EDTA, the percentage of degradation was higher for both dyes in comparison to the use of ZnS nanoparticles without any scavenger. On the one hand, these results suggest that h^+^ has a minor role on the photodegradation process, e^-^ being mainly responsible in forming the oxidizing species. On the other hand, the higher percentage of degradation using the h^+^ scavenger may show that the recombination of electron-holes in these ZnS nanoparticles is higher. Then, the use of h^+^ scavenger reduces the electron-hole recombination, increasing e^-^ which can migrate to the surface of the ZnS nanoparticle and form oxidizing species to degrade the organic dyes.

To analyze the kinetics of the degradation of the organic dyes by ZnS nanoparticles, the experimental results were fitted to the first order kinetic model:(2)$$\ln\frac{C_{0}}{C_{t}}=kt$$

*C*_0_ being the initial absorbance at 665 nm for MB and 555 nm for RhB, *C_t_* the absorbance at time *t* for the same wavelengths, *k* the degradation rate value, and *t* the time. From this model, the *k* obtained rate values were 0.0040 ± 0.0001 min^−1^ and 0.0059 ± 0.0003 min^−1^ for MB and RhB, respectively. As has been amply reported, the degradation rate values of organic dyes in water using a photocatalyst depend on several factor such as initial concentration of organic dyes, the amount of photocatalyst used, temperature and pH, among others [3]. Table 1 summarizes previous results about the photodegradation of organic dyes using ZnS nanostructures, order to compare with these experimental results. It can be observed that other kind of ZnS nanostructures reported higher *k* rate degradation for MB and RhB in water. Nevertheless, these nanostructures were synthesized by chemical-based methods such as precipitation or hydrothermal techniques that use hazardous chemical reagents for synthesis, and generate chemical waste that is difficult to treat. The photocatalytic activity presented for these ZnS nanoparticles provides the opportunity to obtain additional valuable products using the sulfidogenic bioreactor. Moreover, some strategies may be followed to enhance photocatalytic activity. The formation of composites by doping the ZnS nanoparticles with metal defects such as Ag, or the incorporation in another matrix such as carbon nanotubes, enhance the photocatalytic activity for the degradation of organic dyes, facilitating the generation and separation process of excited electron-hole pairs to begin the production of oxidating species [45]. On the other hand, other authors reported an increase in degradation rate by the formation of a ZnS-based dual semiconductor [46,47]. One of the main problems which reduces the photocatalytic activity of ZnS nanoparticles is the rapid recombination of the excited carriers. In this work, the large size and the low stability of the ZnS nanoparticles increased this recombination, reducing photocatalytic efficiency. The formation of dual semiconductor ZnS/CdS or ZnS/CuS resulted in a rise in the photodegradation rate due to the transfer of photogenerated carriers between the semiconductor energy bands, and may improve separation efficiency and reduce the recombination rate [46,47]. These kinds of techniques could be implemented in the synthesis method using sulfidogenic bioreactors; however, further studies should be done.

### 3.3. Antibacterial Activity

The antibacterial activity of ZnS nanoparticles was studied by the well-diffusion method against different bacterial strains. Figure 8a,b shows pictures of the agar plates in diffusion method test of ZnS nanoparticles against *Staphylococcus aureus* using two different concentrations of ZnS nanoparticles, 20 mg/mL and 40 mg/mL. The presence of an inhibition zone around the well can be clearly observed for both concentrations, confirming the antibacterial activity of the ZnS nanoparticles against this strain.

Table 2 summarizes the inhibition zones of both concentration of ZnS nanoparticles against different pathogens. The antimicrobial activity was analyzed against eight different pathogens, four Gram-negative bacteria: *Escherichia coli, Enterobacter cloacae, Klebsiella pneumoniae* and *Pseudomonas aeruginosa,* and four Gram-positive bacteria: *Staphylococcus aureus, Bacillus sp, Enterococcus faecalis* and *Staphylococcus saprophyticus.* The antibacterial activity of ZnS nanoparticles was diverse among the different studied bacteria strain. The greatest inhibition zones were obtained for *E. coli* and *S. aureus* strains. For both strains, the higher the concentration of ZnS nanoparticles, the bigger the inhibition zone, increasing from 17.6 ± 0.6 mm with 20 mg/mL of ZnS nanoparticles to 21 ± 1 mm with 40 mg/mL, and from 14 ± 1 mm to 21 ± 1 mm for the same concentrations, for *E. coli* and *S. aureus* strains, respectively.

Against *K. pneumoniae, Bacillus* sp. and *S. saprophyticus* strains, the ZnS nanoparticles only exhibited antimicrobial activity for the highest concentration (40 mg/mL), while *E. cloacae*, *E. faecalis*, and *P. aeruginosa* strains there was no change in growth for both concentrations.

Other kinds of ZnS nanostructures synthetized by chemical and green synthesis methods have also shown efficient antibacterial activity against *E.coli* and *S. aureus*, exhibiting similar inhibition zones using similar concentrations [37,51,52]_._ However, the main mechanisms involved in the antibacterial effect of ZnS nanoparticles remain unclear, and further studies should be performed. The main mechanisms seem to be the release of metal ions from the nanoparticles and the interaction of the nanoparticles with the surface of bacteria [53]. The interaction of the anions of the cell membrane and cations delivered by the ZnS nanoparticles can create a cellular imbalance, causing cell lysis and bacterial death, specifically the union with the functional groups thiol (SH-), amino (NH_2_^-^) and/or hydroxyl group (OH-) of Gram-negative bacteria. On the other hand, the high surface area and surface chemical reactivity of nanoparticles in comparison to micro and macro particles could cause the adhesion of ZnS nanoparticles to the surface of the bacteria, blocking the exchange energy between the bacteria and the environment, resulting in cell death. According with the experimental results, ZnS bioproduced using H_2_S generated by the sulfidogenic bioreactor seems to have higher antibacterial activity against Gram-positive bacteria rather than Gram-negative bacteria, similar to previous works [53]. However, contradictory results have also been reported in which ZnS nanoparticles fabricated by a green synthesis method were able to inhibit the bacterial growth of *P. aeruginosa* in contrast with the present results [51]. However, residues of the plant extract used in the synthesis and present on the surface of ZnS nanoparticles could be responsible for the better antibacterial activity found in that work [51].

## 4. Conclusions

ZnS nanoparticles were successfully synthesized using green H_2_S gas flux produced by a sulfate reducing bioreactor. ZnS nanostructures showed a spherical-like shape with a mean diameter of 107 nm, with principally a zinc-blende crystalline structure and a main crystalline size of 4.7 nm. Moreover, they showed semiconductor behavior with an electronic band gap of 3.73 eV and a pure ZnS chemical structure according to XPS measurements. In addition, these ZnS nanoparticles exhibited multifunctional applications including photocatalytic activity for the degradation of MB and RhB in water under UV radiation, and antibacterial activity against several bacterial strains including *Escherichia coli, Klebsiella pneumoniae, Staphylococcus aureus* and *Staphylococcus saprophyticus*. Our results offer a new synthesis method to obtain valorous ZnS nanostructures by using a sulfidogenic bioreactor, which could be used for water treatment and on antibacterial surfaces.

## Figures and Tables

**Figure 1 nanomaterials-13-00935-f001:**
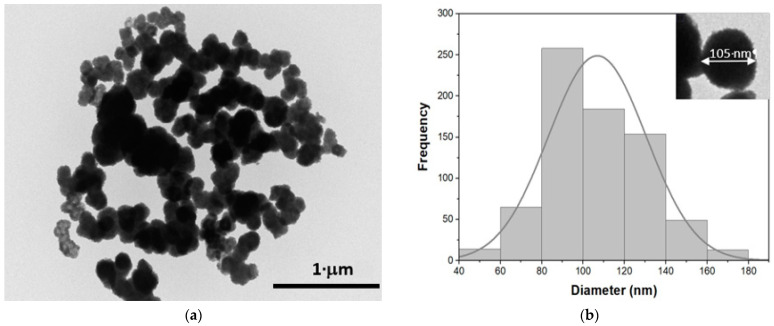
(**a**) Transmission electron microscopy (TEM) image of ZnS nanoparticles. (**b**) Histogram of size distribution of ZnS nanoparticles. Inset a zoom of a TEM image showing the spherical-like shape of a ZnS nanoparticle with a mean diameter of 105 nm.

**Figure 2 nanomaterials-13-00935-f002:**
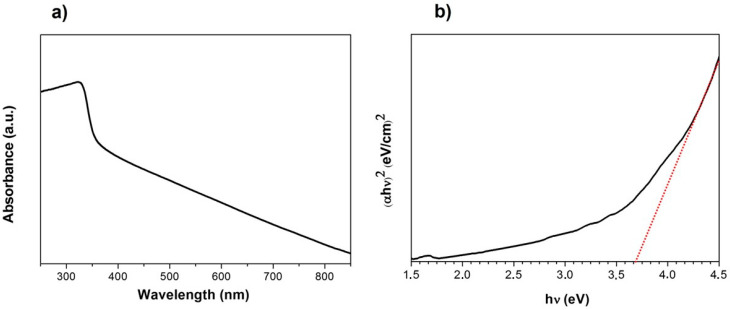
(**a**) UV-vis absorption spectrum of ZnS nanoparticles. (**b**) Tauc Plot obtained from UV-vis absorption spectrum.

**Figure 3 nanomaterials-13-00935-f003:**
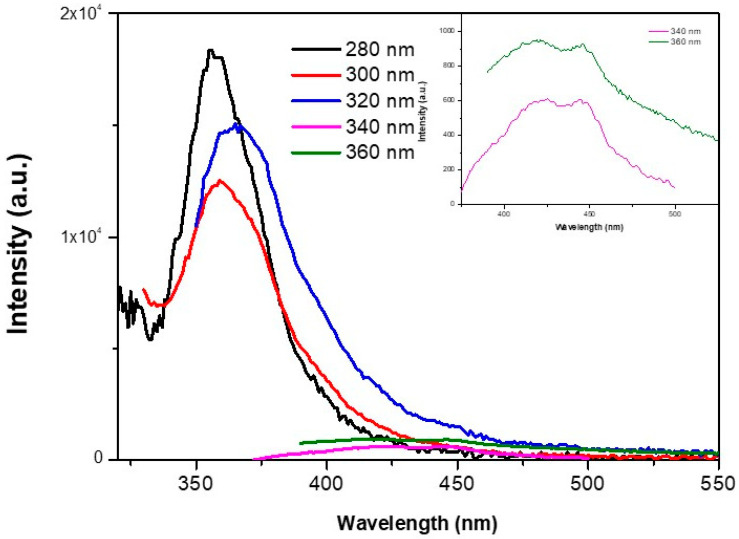
Fluorescence emission spectra of ZnS nanoparticles with different excitation wavelengths. Inset, a zoom of the fluorescence spectra with 340 nm and 360 nm excitation wavelengths.

**Figure 4 nanomaterials-13-00935-f004:**
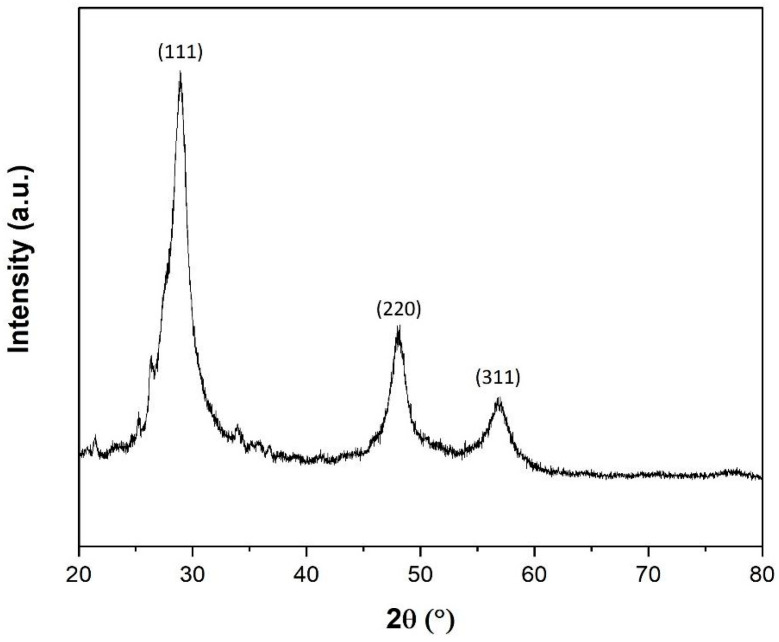
XRD pattern of ZnS nanostructures.

**Figure 5 nanomaterials-13-00935-f005:**
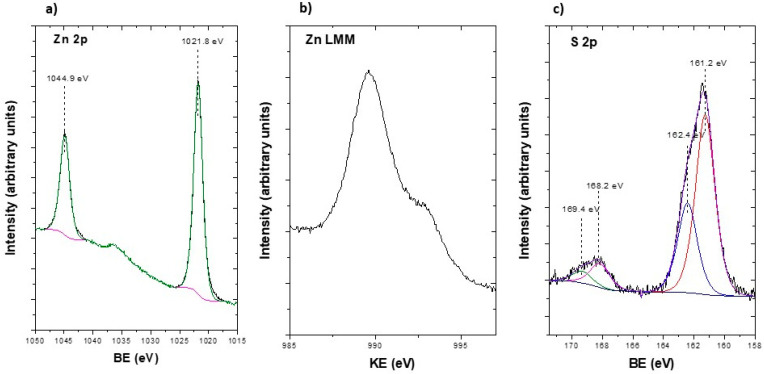
(**a**) Zn 2p (**b**) Zn LMM Auger transition and (**c**) S 2p XPS spectra of ZnS nanoparticles, respectively.

**Figure 6 nanomaterials-13-00935-f006:**
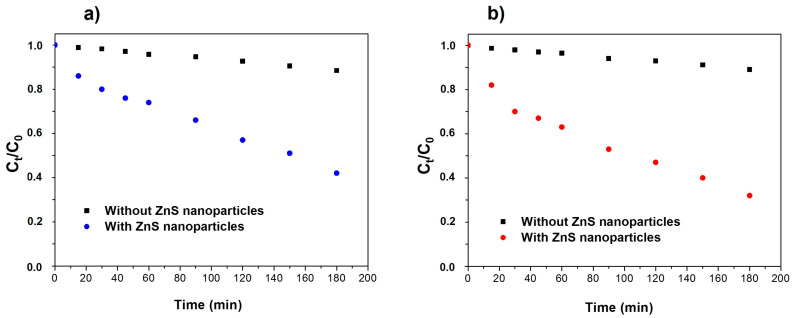
Photocatalytic degradation of (**a**) methylene blue and (**b**) rodhamine_B with and without a ZnS photocatalyst irradiated with UV light.

**Figure 7 nanomaterials-13-00935-f007:**
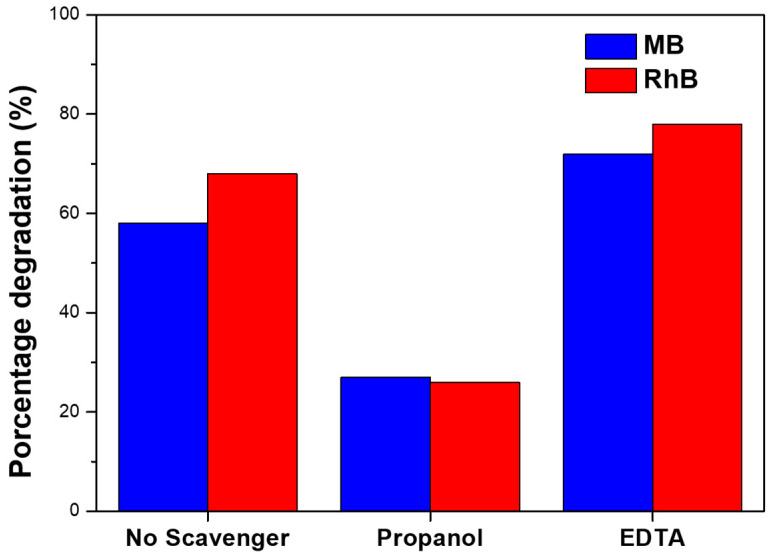
Effects of scavenger on MB and RhB photodegradation by ZnS nanoparticles.

**Figure 8 nanomaterials-13-00935-f008:**
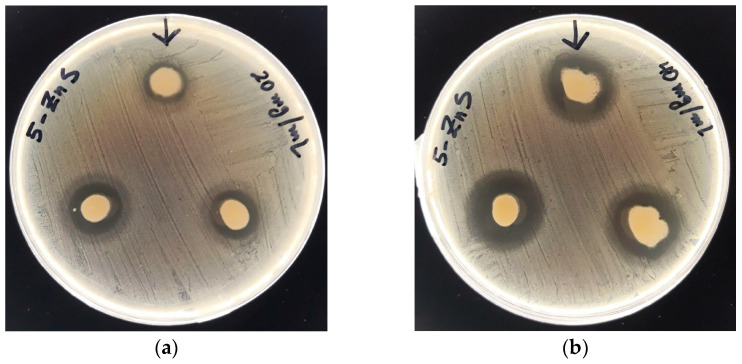
Inhibition zone for *Staphylococcus aureus* strain in the presence of ZnS nanoparticles at two concentrations (**a**) 20 mg/mL and (**b**) 40 mg/mL.

**Table 1 nanomaterials-13-00935-t001:** Previously reported results of photocatalytic activity of ZnS nanostructures.

Sample	Synthesis Method	Size (nm)	Dye Concentration	Photocatalyst	K Rate (min^−1^)	Reference
ZnS	Coprecipitation	40	MB-50 mg/L	0.5 mg/mL	0.02	[10]
ZnS	Lyotropic Liquid crystal template	60–200	MB-20 mg/L	1 mg/mL	0.04	[48]
ZnS	Green-synthesis	11.8	MB-10 mg/L	5 mg/mL	0.002	[8]
ZnS-PVDF	Coprecipitation	90	MB 10 mg/L	5 mg/mL	0.0163	[49]
ZnS	Sulfidogenic bioreactor	107	MB 10 mg/L	1 mg/mL	0.004	This work
ZnS-Ag	Sonochemical method	15	RhB 10 mg/L	1 mg/mL	0.25	[45]
ZnS	Ion exchange method	>10	RhB 4.8 mg/mL	1 mg/mL	0.0008	[46]
SnO_2_-ZnS	Hydrothermal	65	RhB 10 mg/L	1 mg/mL	0.046	[47]
ZnS–ZnFe_2_O_4_	Hydrothermal	100–200	RhB 20 mg/L	0.6 mg/mL	0.0031	[50]
ZnS	Sulfidogenic bioreactor	107	RhB 10 mg/L	1 mg/mL	0.0059	This work

**Table 2 nanomaterials-13-00935-t002:** Antibacterial inhibition zone (mm) of ZnS nanoparticles.

ZnS Concentration	Inhibition Zone (mm)							
	*E. coli*	*E. cloacae*	*K. pneumoniae*	*P. aeruginosa*	*S. aureus*	*Bacillus* sp.	*E. faecalis*	*S. saprophyticus*
20 mg/mL	17.5 ± 0.5	0.0 ± 0.0	0.0 ±0.0	0.0 ± 0.0	14 ± 1	0.0 ± 0.0	0.0 ± 0.0	0.0 ± 0.0
40 mg/mL	21 ± 1	0.0 ± 0.0	14.0 ± 0.7	0.0 ± 0.0	21 ± 1	10.4 ± 0.6	0.0 ± 0.0	16 ± 2

## Data Availability

Not applicable.

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
