# Peer review of "Sulfidogenic Bioreactor-Mediated Formation of ZnS Nanoparticles with Antimicrobial and Photocatalytic Activity"

_nanomaterials, 2023, doi:10.3390/nano13050935_

Round 1

Reviewer 1 Report (Previous Reviewer 1)

The manuscript can be reconsidered after its revision. Below are comments to help the authors improve it:

1\ Fig.5, legends to X-axis: it's better to spell out both BE and KE, many readers may not know the meaning of these abbreviations

2\ Line 293: NH2-  must be presented correctly

3\ Fig.4, caption: probably, there is only ONE pattern, not "patterns" here? Please check

4\ Line 471: ZnFe2O4 must be presented correctly

5\ Line 363: ZnS must be presented correctly

6\ Lines 340 & 350: SnS2 must be presented correctly

7\ Lines 324-325: the journal title must be abbreviated correctly and properly

8\ Lines 371 & 386-387: the journal title must be abbreviated correctly and properly

9\ Lines 393-394: the journal title must be abbreviated correctly and properly

10\ Lines 430 & 475: the journal title must be abbreviated correctly and properly

Author Response

The manuscript can be reconsidered after its revision. Below are comments to help the authors improve it:

1\ Fig.5, legends to X-axis: it's better to spell out both BE and KE, many readers may not know the meaning of these abbreviations

R: X-axis of Figure 5 were changed: BE was renamed as Binding energy in Figures 5a and 5c, and KE was renamed as Kinetic energy in Figure 5b.

2\ Line 293: NH2-  must be presented correctly

R: We have written NH2- correctly (Line 296)

3\ Fig.4, caption: probably, there is only ONE pattern, not "patterns" here? Please check

R: We have change Fig.4, caption after reviewing it. As the reviewer suggested, there is only one pattern

4\ Line 471: ZnFe2O4 must be presented correctly

R: We have written ZnFe2O4 correctly (Line 474)

5\ Line 363: ZnS must be presented correctly

R: We have written ZnS correctly (Line 366)

6\ Lines 340 & 350: SnS2 must be presented correctly

R: We have written SnS2 correctly (Line 343 & 353)

7\ Lines 324-325: the journal title must be abbreviated correctly and properly

R: We have abbreviated the name of the journal (line 328)

8\ Lines 371 & 386-387: the journal title must be abbreviated correctly and properly

R: We have abbreviated the name of the journal (line 374 & 389)

9\ Lines 393-394: the journal title must be abbreviated correctly and properly

R: We have abbreviated the name of the journal (line 396-397)

10\ Lines 430 & 475: the journal title must be abbreviated correctly and properly

R: We have abbreviated the name of the journal (line 433 & 478)

Reviewer 2 Report (Previous Reviewer 3)

The authors have solved all the issues. I recommend accepting this article.

Author Response

The authors have solved all the issues. I recommend accepting this article.

R: Thank you for your recommendation

Reviewer 3 Report (New Reviewer)

The manuscript by A. Segura et al., reports on the antimicrobial and photocatalytic activity of ZnS nanoparticles prepared by a sulfidogenic bioreactor. They use various diffraction and optical techniques to study these NPs, which reveal spherical shapes possessing zinc-blende crystal structures. I find this work interesting. However, I have the following concerns and suggestions, which I believe will be helpful in improving the manuscript.

First of all, the specific properties of ZnS and other similar applications can be shortly emphasized in the introduction. I hope the following papers will be helpful. 

https://doi.org/10.1016/j.jmat.2019.02.002

https://doi.org/10.1016/j.cplett.2016.01.009

https://doi.org/10.1016/j.apcatb.2019.04.008 https://doi.org/10.1016/j.matlet.2015.09.108

I suggest the authors compare the crystalline size obtained with the XRD and nanoparticles size from TEM in order to differentiate between NPs and crystalline domains.

In lines 183-184, the authors mention that "This result is in agree with the calculated optical band gap." It stands abstract what the authors want to convay.

In general, the text has some flow and construction problems. For example, it would be better if the application (i.e., catalytic test, antibacterial test) is discussed in a separate section. 

In table 1, the catalytic activities of ZnS are compared with the available literature. Although it is useful data, I think these outcomes are not directly comparable since using different scavengers also affects the results, as also stated by the authors on page 9.

There are some typos and grammatical errors that must also be corrected in this revision process. Such as, "like-spherical shape" should read "spherical-like shape" or just "sphere shape". 

In page 13, line 293-294, the authors mention "high surface area of the obtained nanoparticles to improve antibacterial capability". In principle, the surface area, which is also an important parameter for photocatalytic performance, can be calculated using Brunauer–Emmett–Teller (BET) measurements. I hope the authors can discuss this further.

One of the main concerns about NPs for application is their stability over time or against special environments. I suggest the authors include discussions regarding stability.

The authors mention NPs size being around 100 nm. Is it possible to reduce the particle sizes with this method?

In general, the manuscript contains useful results. Therefore, I suggest the authors elaborate on my comments and correct typos, errors, etc., before the final decision. 

Author Response

The manuscript by A. Segura et al., reports on the antimicrobial and photocatalytic activity of ZnS nanoparticles prepared by a sulfidogenic bioreactor. They use various diffraction and optical techniques to study these NPs, which reveal spherical shapes possessing zinc-blende crystal structures. I find this work interesting. However, I have the following concerns and suggestions, which I believe will be helpful in improving the manuscript.

First of all, the specific properties of ZnS and other similar applications can be shortly emphasized in the introduction. I hope the following papers will be helpful. 

https://doi.org/10.1016/j.jmat.2019.02.002

https://doi.org/10.1016/j.cplett.2016.01.009

https://doi.org/10.1016/j.apcatb.2019.04.008

https://doi.org/10.1016/j.matlet.2015.09.108

R: Thank you for the suggestion. ZnS properties and applications are emphasized in the introduction (lines 43-52), with references [4-10]. We think this information is enough for the purpose of the manuscript.

I suggest the authors compare the crystalline size obtained with the XRD and nanoparticles size from TEM in order to differentiate between NPs and crystalline domains.

R: Following the reviewer suggestion, we have improved the text to emphasis this concept in lines 181-185. “The obtained average crystalline domain was D = 4.7 ± 0.4 nm.  The relative small magnitude of the crystalline size in comparison with the size of the ZnS nanoparticles obtained by TEM (107 ± 20 nm) confirms the mixture of both crystalline structures inside the NPs, being the zincblende the predominant structure.”

In lines 183-184, the authors mention that "This result is in agree with the calculated optical band gap." It stands abstract what the authors want to convay.

R: We have improved the sentence for a better understanding in lines 185-186. “This result is in agree with the calculated optical band gap which matches with the values previously reported for ZnS nanoparticles with a mixed of crystalline structures [30,31]”

In general, the text has some flow and construction problems. For example, it would be better if the application (i.e., catalytic test, antibacterial test) is discussed in a separate section. 

R: In order to improve the structure of the manuscript, we have divided section 3 in three different subsections, which are 3.1. ZnS NPs characterization (page 4, line 127)   3.2. Photocatalytic activity (page 8, line 205) and 3.3 Antibacterial activity (page 11, line 261)

In table 1, the catalytic activities of ZnS are compared with the available literature. Although it is useful data, I think these outcomes are not directly comparable since using different scavengers also affects the results, as also stated by the authors on page 9.

R: The photodegradation of organic dyes using ZnS nanostructures present in Table 1 report k rate degradation for MB and RhB in water. Although we do not achieve the higher k rate we think it is important to show the state of the art of this parameter

There are some typos and grammatical errors that must also be corrected in this revision process. Such as, "like-spherical shape" should read "spherical-like shape" or just "sphere shape". 

R: Thank you for your comments. We have checked and typos and grammatical errors in order to improve the manuscript

In page 13, line 293-294, the authors mention "high surface area of the obtained nanoparticles to improve antibacterial capability". In principle, the surface area, which is also an important parameter for photocatalytic performance, can be calculated using Brunauer–Emmett–Teller (BET) measurements. I hope the authors can discuss this further.

R: We agree with the reviewer that surface area is an important parameter to explain the antibacterial and photocatalytic activity of nanoparticles, specially when comparing to different kind of nanoparticles with different degradation rates or antibacterial activities, and this parameter can be obtained by BET measurements. In this work, we have just studied one kind of nanoparticle, and the BET value will not give so important information.

In line 293, we refer that ZnS nanoparticles have higher surface area than ZnS micro and macroparticles, and this can explain why ZnS exhibit antibacterial activity in contrast to ZnS macroparticles. In order to explain that in details, we have changed the text at page 13 line 295:On the other hand, the high surface area and surface chemical reactivity of nanoparticles in comparison to micro and macro particles

One of the main concerns about NPs for application is their stability over time or against special environments. I suggest the authors include discussions regarding stability.

R: We agree that stability is an important parameter for using nanoparticles in several applications. In this work, ZnS nanoparticles were stored at standard conditions for temperature and pressure after they were collected for the solution as it is explained in the materials and method section, and they exhibited similar photocatalytic and antibacterial activities independently of the time of store. However, further studies should be performed to research the stability of these nanoparticles in order ambient as the reviewer suggests. 

The authors mention NPs size being around 100 nm. Is it possible to reduce the particle sizes with this method?

R: By this method, the control of the size of the ZnS precipitates is complicated and it depends on several parameters including H2S flux, Zn concentration, temperature, and so on. For higher H2S flow, ZnS micro and macroparticles are formed. At this low flux, we achieve ZnS nanoparticles around 100 nm of diameter. It is possible that with lower flux, or lower Zn concentration we could achieve ZnS nanoparticles with smaller size. Thank you for your comment which can be a point of start for further experiments.  

In general, the manuscript contains useful results. Therefore, I suggest the authors elaborate on my comments and correct typos, errors, etc., before the final decision. 

R: Thank you for your comments. We have considered all your comments and we think the manuscript has been improved

Round 2

Reviewer 1 Report (Previous Reviewer 1)

The manuscript was improved and thus can be accepted

Author Response

Thank you for your comments

Reviewer 3 Report (New Reviewer)

The authors have partially responded to the reviewer's reports. I followed their corrections to the manuscript, which I assume reproduced in red. In general, it is inadequate. The manuscript still suffers from back-and-force statements, poor discussions, and flow. Data evaluation and presentation have severe problems. I suggest the authors recheck their results, particularly in Fig 2, Fig 3, and Fig 5, and evaluate the data carefully. Also, for the result discussions, consider all the references suggested by the reviewers and include proper comparisons. 

Author Response

The authors thank the comments of the referees which have improved the quality of the manuscript. Here, you can find a point by point the answer to your suggestions, which are highlight in red in the revised manuscript. 

New experiments have been carried out to recheck the data of figure 2, and similar results have been obtained. The new data have been included in the new figure 2.  More details about the performance of the experiment have been included in the material and method section.  Moreover, the results have been revised and a deeper discussion has been included in the text, including comparations with the references suggested by the referee at page 5, line 141: “Figure 2a shows UV-vis absorption spectrum of ZnS nanoparticles. A broad absorption band from 250 to 350 nm of wavelength is observed with the maximum absorption peak at 335 nm of wavelength. Then, the absorbance decays over higher wavelengths. The shape of absorption spectrum is blue-shifted with respect to the reported for ZnS macroparticles, due to the quantum confinement effects [27]. These effects have been also reported for other kinds of ZnS nanoparticles. ZnS nanoparticles with smaller diameters and synthesized by layer by layer in a polymeric matrix showed similar absorption bands in the same range, presenting alike blue shifts which depend on the size of the nanoparticle [28]. Similarly, ZnS nanoparticles formed by co-precipitation method using Schiff base exhibited absorbance spectrum with similar effects [29]. On the other hand, monodisperse ZnS nanoparticles can show defined absorption peaks correlated to the quantum effects as it was reported on ZnS nanoparticles synthesized via chemical in situ techniques [30,31]. Moreover, the quantum effects may be influenced by the morphology of the ZnS nanoparticles and its stability [32]. In this work, the not-well defined absorption band can be attributed to nanoparticles morphology characterized by sizes higher than 100 nm, low stability and tendency to agglomerate [31,32]. From UV-vis spectra, the optical band gap was calculated from Tauc plot (Figure 2b). For that, first the absorption coefficient (α) was shifted such that the lowest value was set to zero to account for any wavelength-independent reflectance and scattering [33,34]. Then, (αhν)2 was plotted as a function of photon energy hν, and the band gap is obtained by extrapolating the linear region to the horizontal axis. In this case, the obtained band gap was 3.73 eV. This value can be modified depending on several factor such as the crystallinity, the shape, the stability, among others. ZnS nanoparticles with lower main diameters usually showed higher energy band gap values [29,30]. ZnS nanoparticles synthesized by chemical route showed little different band gaps, ranging between 3,98 and 3.87 eV depending on the reaction time [29]. On the other hand, ZnS nanostructures with diverse morphologies can modified the band gaps from 3.5 to 3.8 eV [32]. The value obtained in this work matches with other kind of nanoparticles with mixed of crystalline structures [32,35].”

Regarding the Figure 3, a comparative study with other works to improve the discussion of the results was added at page 6, lie 170: “The photoluminescence is an important optical parameter for the development of optoelectronic and sensing applications, which is influenced by the shape, the size and the surface and effect energetic states, among others. Figure 3 shows the fluorescence spectra of ZnS using different excitation wavelengths. For excitation wavelengths with higher energy than the band gap (280 nm, 300 nm and 320 nm), an emission peak can be observed centered around 355 nm and 365 nm. These emission peaks may be related to the recombination of electron-hole pairs after the photon absorption, being similar than other previously reports  [36,37]. Moreover, the broad peak can be associated to the wide size distribution of the ZnS nanoparticles since the quantum effects of the nanometer size can derive in different wavelength emission. In addition, a small red shift of these peaks is observed as the excitation wavelength is increased. The one obtained at an excitation of 280 nm is centered at 356 nm and the peak is shifted to 360 nm and 366 nm for 300 nm and 320 nm of excitation wavelength, respectively. This shift can be related to selective excitation of the surface states in the nanoparticles [38].  For higher excitation wavelength (340 nm and 360 nm), this main peak disappears. However, others smooth peaks can be observed (inset Figure 3) at 427 nm and 445 nm. These peaks can be related to defect states or interstitial impurities [30,36]. Nonetheless, the main fluorescence emission is presented over the UV range, being difficult to be used for optoelectronic or sensing applications [31]. Other well defined ZnS nanoparticles synthesized by chemical methods could be shifted the photoluminescence emission to the visible wavelength range [29].  Balayeva N. et al observed an emission band centered at 430 nm for ZnS nanoparticles with a main diameter of 10 nm [28],  and Mamiyev Z. et al reported emission bands ranged from 401 nm to 418 nm, depending on the main size of each nanoparticle [29] . In addition, Schiff base coordinated ZnS nanoparticles excited at 340 nm, exhibited a broad photoluminescence emission centered at 432 nm [30]. These effects were explained by electron-hole recombination with energy states within the band gap. These energy states can be associated to surface defects, defect states provoked by S and Zn vacancies, and the size dispersion of the nanoparticles [29]  ”

Figure 5 has been also revised, and additional information was added to complete the discussion.

Moreover, the conclusions were adapted to the new discussions and new references were added.

  1. Jang, J.S.; Yu, C.J.; Choi, S.H.; Ji, S.M.; Kim, E.S.; Lee, J.S. Topotactic Synthesis of Mesoporous ZnS and ZnO Nanoplates and Their Photocatalytic Activity. J Catal 2008, 254, 144–155, doi:10.1016/j.jcat.2007.12.010.
  2. Balayeva, N.O.; Mamiyev, Z.Q. Synthesis and Studies of CdS and ZnS-PE/NBR Modified Thermoplastic Elastomeric Copolymer Nanocomposite Films. Mater Lett 2016, 162, 121–125, doi:10.1016/j.matlet.2015.09.108.
  3. Mamiyev, Z.Q.; Balayeva, N.O. Optical and Structural Studies of ZnS Nanoparticles Synthesized via Chemical in Situ Technique. Chem Phys Lett 2016, 646, 69–74, doi:10.1016/j.cplett.2016.01.009.
  4. Ayodhya, D.; Veerabhadram, G. Fabrication of Schiff Base Coordinated ZnS Nanoparticles for Enhanced Photocatalytic Degradation of Chlorpyrifos Pesticide and Detection of Heavy Metal Ions. Journal of Materiomics 2019, 5, 446–454, doi:10.1016/j.jmat.2019.02.002.
  5. Madhusudan, P.; Wang, Y.; Chandrashekar, B.N.; Wang, W.; Wang, J.; Miao, J.; Shi, R.; Liang, Y.; Mi, G.; Cheng, C. Nature Inspired ZnO/ZnS Nanobranch-like Composites, Decorated with Cu (OH) 2 Clusters for Enhanced Visible-Light Photocatalytic Hydrogen Evolution. Appl Catal B 2019, 253, 379–390.

This manuscript is a resubmission of an earlier submission. The following is a list of the peer review reports and author responses from that submission.

Round 1

Reviewer 1 Report

The manuscript needs revision. Below are comments and suggestions to help the authors improve it:

1) Please check the title: probably , the "the" should be removed between "mediated" and "formation".

2) English should be improved. Below are just several example, while there are many more errors in the manuscript:

2.1) Abstract, line 4 from bottom:  Methylene blue and Rhodamine B are not personal names and do not need any capital letters . Compare page 2, line 5 from top

2.2) page 1, line 3 from bottom:   ZnS nanostructures ... due to its (due to their?)

2.3) page 2, lines 19-20 from top: ZnS nanoparticles were .... its photocatalytic activity (their?)

2.4) page 2, line 13 from bottom: there "of" between 5 mM and ZnSO4 should be deleted

2.5) page 3, line 3 from bottom: similarly than other kind of  (this phrase is not readable)

2.6) page 4, line 6 from bottom:  ... two main peak (peaks?)

3) page 2, line 7 from top:  must be [5-7]

4) page 3, line 11 from top: there is no need to introduce MB and RD here, these acronyms were already introduced on page 2 (line 5 from top)

5) page 3, line 10 from bottom: "24 hours" should be replaced with "24 h"

6) page 4, Fig.1a: the scale bar is not well seen. Please enlarge the numbers/values that describe the scale bar

7) page 7, Fig.5 ,caption:   What kind of Zn nanoparticles was prepared in the present manuscript? Did the authors really prepare metallic nanoparticles?

8) page 2, paragraph 2 from top: when mentioning and discussing methods previously used to prepare ZnS nanoparticles, the authors are recommended to mention (and discussed) more methods and reports: 

Carbohydrate Polymers  2022, 288, 119332

Environmental Research 2020, 186, 109513

Materials 2020, 13, 171

Open Chemistry 2021, 19, 1134-1147

Materials 2019, 12, 3313

9) page 9, Table 1, column 4: what is the meaning of hyphen in this column?  Mb-50mg/L ; MB-20mg/L; MB-10mg/L

10) page 13, ref.13: "H2S" must be presented correctly (with subscript)

Reviewer 2 Report

  1. The background of formation of ZnS should be added in the Introduction.
  2. How could the author obtain that the ZnS nanoparticles present a spherical-like shape by the TEM showed in Figure 3.1?
  3. In Figure 2b, the red line is not the extrapolating of the linear region, thus, the band gap is wrong?
  4. The peaks in Figure 3 are all smooth.
  5. The photo-degradation experiments of Methylene blue and Rhodamine_B should be discussed in detail.

Reviewer 3 Report

The authors reported the ZnS photocatalyst obtained by using the “green” H2S gas produced by a sulfidogenic bioreactor. ZnS nanoparticles were able to degrade MB and RhB in water under UV radiation, and also showed high antibacterial activity against different bacterial strains including Escherichia coli and Staphylococcus aureus. However, there are a number of major problems with the manuscript. Unfortunately, I recommend that it be rejected for further treatment.

  1. ZnS with a 3.68 eV band gap does not have any photocatalytic application value, although it comes from the "green" H2S gas produced by a sulfidogenic bioreactor. Ultraviolet light, which absorbs only a small proportion of solar energy, greatly limits its use. Its photocatalytic performance for degradation of dyes MB and RhB may be due to the sensitization of dyes in the reaction system.
  2. The importance and innovation of this work are not seen from the introduction part.
  3. The fluorescence spectra in the paper are apparently strange, possibly due to unsuitable test conditions, and the authors do not explain why this occurs.
  4. In the dye degradation diagram (Fig. 6), so many sample test points would cause great errors in the whole reaction system. In addition, the degradation effect was also poor, with only half of the dye being degraded within 3 hours.
  5. There are some small problems, such as “Zn nanoparticles” should be changed to “ZnS nanoparticles” in Fig. 5, “RD” should be changed to “RhB”.